# Can high-performance work practices influence employee career competencies? There is a need for better employee outcomes in the banking industry

**Damis Feruzi Kamna**[1]*, **Shiva Ilkhanizadeh**[2]

**1** Department of Business Administration, Cyprus International University, Lefkosa, Turkey, **2** School of Tourism and Hotel Management, Cyprus International University, Lefkosa, Turkey

* damiskamna@hotmail.com

**Data Availability Statement:** All relevant data are within the manuscript.

**Funding:** The authors received funding from the Institute of Public Administration in Zanzibar,

## Abstract

It is essential for organizations to invest and improve employee outcomes to enhance organizational competitiveness and growth in today's world. However, most organizations place management objectives above the career competencies of employees. Therefore, this study investigated 1. the effect of high-performance work practices on employee career competencies in the banking industry. 2. the mediating effect of employee career competencies on the relationship between high-performance work practices and employee outcomes in the banking sector. The study adopted a quantitative approach with a total of 340 respondents from various banks in Tanzania. The data was analyzed using Covariance Based Structural Equation Modelling (CB-SEM). The results of the finding indicate that high-performance work practices have a significant effect on employee career competencies. Similarly, employee career competencies significantly impact service quality, creative performance, and extra-role performance in banks. Also, employee career competency does not mediate the relationship between high-performance work systems and service quality in the banking industry.

## Introduction

Strategic human resource management (SHRM) has researchers assessing the conceivable share of using high-performance work systems (HPWSs) as a method to expand an organizations' competitive advantage [1–4]. HPWS refers to an interconnected group of HR policies and practices that involves thorough selective staffing, general training and development, incentive rewards, good work-life balance, adequate empowerment, job security and, great career opportunities, which are calculated to improve employees' career competencies and achieve organizational objectives [1, 5–9]. Although, the preceding HPWS study has displayed a relationship between employee well-being and organization performance [1, 10, 11]. The new HPWS research is often critiqued for its extensive management-centric viewpoint, prioritizing objectives over employee well-being [12]. This viewpoint is apparent in its limited

Tanzania, which provides full financial support for the study. The funders had no role in study design, data collection and analysis, decision to publish, or preparation of the manuscript.

**Competing interests:** The author(s) received no specific funding for this work.

economic outlook of HPWS research, which centers around improving the competence of HR practices [13, 14], while employee's effects on HPWS have not been given the appropriate attention [15].

As a result, some researchers have requested studies centered on employee research and also features the managerial and functionalist viewpoints, which is a well-adjusted tactic that seeks to evade the sidelining of employee outcomes in HPWS [12, 16, 17]. Lepack et al. [8] also stated that the HR systems should be aimed at organizations' tactical objectives and should function well by managing employees at work, especially planning and handling their careers [18]. Arguable, human factors are the most esteemed factors in modern organizations, and offering employees a long-term secure career is a win-win position for organizations and their employees. Szabó-Bálint [19] described a career as a lasting course, which comprises a series of actions, mindsets, or behaviors in a person's work life. Career development relates to the development of employees that is deemed advantageous to both the individual and organization [20], which also encompasses a structured, official, and well-calculated effort to accomplish equilibrium between the individual's career prerequisites and the organization's workforce requests. Employees must take responsibility for their employability by acquiring experience and career competencies.

To maintain a job in this dynamic labor market, individuals need progressive career competencies that can help them in their careers [21]. When an organization provides the appropriate HPW practices, employees discover that their job competencies meet the organization's standards; they would likely exhibit employee outcomes such as service quality, creative performance, and extra-role behavior. No study has been able to propose this conceptualization. Therefore, a conceptual framework was developed to examines the effects of high-performance work practices on career competencies. Furthermore, the career competencies relationship between service quality, creative performance, and extra-role behavior was examined. Lastly, the career competencies construct is examined as a mediator between service quality, creative performance, and extra-role behavior.

## Literature reviews

### HPWPs and career competencies

This study focuses more on employee-driven HPWP, which has not received enough attention. Studies relating to the effect of HPWP on employees' pro-organizational behavior are habitually grounded on the social exchange theory (SET). This denotes that employees feel responsible for responding to the organization's benefits when they receive favorable treatments from organizations [22]. Social exchange connections spur affirmative job or organizational conclusions when the organization cares for its employees [23]. Employees that attain economic and socio-emotional capabilities in a workspace are usually compelled to reimburse the organization in several ways. In the absence of specific tactical goals within an organization, the HR work systems would not lead the employees. An organization must select and design the appropriate HR practices that are not only allied with each other but also affiliated with the organization's plan to accomplish the suitable organizational outcomes [1, 10], such as several components of HPWSs and the assessment of social responsibility in employment relations [24–26]. This can be stated that HPWPs are exhibited through the organization's prominence on these components. Organizations that partake in HPWPs often foster motivation amongst their employees, resulting in positive outcomes for organizations [26]. There is a growing need for employees to manage their careers actively; it is gradually becoming essential for them to acquire the crucial competencies that empower them to succeed [27]. Therefore, they must become prolific in the career-related competencies that can help them steer through

their career. Career competencies concern an individual's entire career and could be distinguished from job skills and work capabilities geared towards effectively executing a job [28]. It can be said that career competencies relate to knowledge, skills, and abilities. Career construction theory positions that persons build their careers by enforcing meaningful experiences to their vocational and occupational performances [29]. Employees should acquire the necessary resources to deal with career loads and pressures [30]. With HPWPs in place, employees relish working for the organization while following their objectives and building their career competencies. This occurs when organizations carefully recruit the appropriate personnel to boost their productivity. According to Kong et al. [31], employee empowerment promotes competence and growth and helps employees reach their goals as long as the organization sticks to training them in line with their qualifications. This increases their capacity and provides them with the opportunity to use their empowerment in their work successfully. Also, encourage them to effectively manage their job requirements while working in an environment where remuneration, career development, and job security are highly considered [32]. Oliinyk [26] stated that HPWPs have a significant relationship with career adaptability. Therefore, we hypothesize that:

*H1*: *HPWPs significantly influence Career competencies.*

## Career competencies on employee outcomes

Career competencies are presumed to play a vital role in upholding the employee's worth to the organization [33]. Service quality, creative performance, and extra-role performance are the three imperative employee outcomes of career competencies tested in this study. This study concentrates on these outcomes because no empirical study has tested them with career competencies. Previous research examined career adaptability in terms of meeting expectations, creative performance, and extra-role performance [26]. Whereas service quality is the process of delivering a service and the value of the service outcome. It is the gap between the expectation of the service and satisfaction derived.

Service quality is a determining factor in the provision of service. It does not only become a source of identity by customers, but it becomes an image of the organization. When employees have the knowledge, skills, and abilities required to carry out their job, they can provide supreme service to their customers [34]. Creative performance is the activity that influences "*products, ideas, or procedures that satisfy two conditions, namely, they are novel or original, and they are potentially relevant for, or useful to an organization*" [35, 36]. An employee that is curious about new ways to carry out their work efficiently and effectively can successfully run their careers [28]. When employees are confident in their careers, they search for opportunities for personnel development by solving problems and gaining new skills [37]. Extra-role performances are employee work behaviors that involve providing services to customers that stretch farther than their formal role requirements. Employees who possess career competencies are willing to go above and beyond the line to satisfy their customers. Organizations prefer to recruit employees who perform extra-role behavior. Due to the limited empirical evidence on these relationships, we hereby propose the following hypothesis:

*H2a*: *Career competency significantly influences service quality.*

*H2b*: *Career competency significantly influences creative performance.*

*H2c*: *Career competency significantly influences extra-role performance.*

## Career competencies as a mediator

Organizations that operate with HR practices motivate the employees to go the extra mile to satisfy the customers and explore new ways to carry on their responsibilities to achieve their goals, which inadvertently boosts service quality. An organization with HPWPs incorporates employees who possess the knowledge, skills, and abilities to provide extra-role and creative performances [26]. This notion gives rise to these hypotheses:

*H3*: *Career competencies significantly influence a relationship between HPWP on (a) service quality, (b) creative performance (c) extra-role performance.*

## Methodology

The research model of this study is depicted in Fig 1. based on the effect of high-performance work practices on employee career competencies and the mediating effect of employee career competencies on the relationship between high-performance work practices and employee outcomes in the banking sector.

### Data collection and sample

This study was conducted according to the guidelines laid down in the Declaration of Helsinki, and the Cyprus International University Review Committee approved all procedures involving human subjects. The hard-copy questionnaire was distributed, and data from this survey were collected from banking employees across Tanzania—main cities of Dar es Salaam, Arusha, Mwanza, Moshi, and Dodoma. Dar es Salaam has 290 branches, followed by Arusha, 68 branches, Mwanza, 67, Moshi, 46, and Dodoma, 41. Available data from the Bank of Tanzania annual report shows that in 2020, banking supervisors and managers in this region have a total of 1,198 employees. Therefore, Yamane [38] sample size calculator was used to establish the minimum sample size for the survey, and the result shows that 301 participants are required for the survey. Also, we ensured that prospective respondents were eligible to participate in

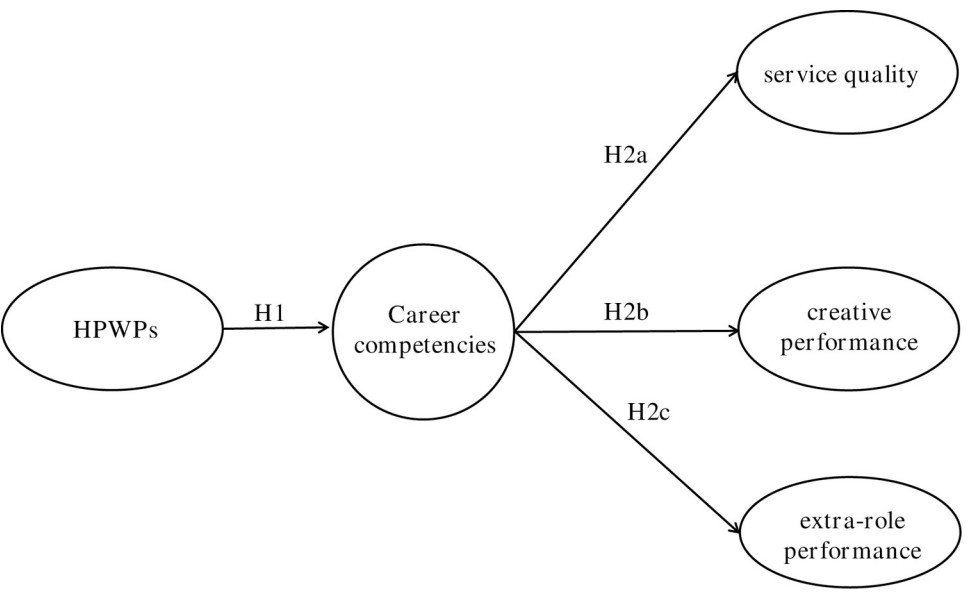

**Fig 1. Conceptual model.**

**Table 1. The demographic characteristics of the sample.**

| | | Frequency | % |
|---|---|---|---|
| **Gender** | Female | 139 | **40.42** |
| | Male | 195 | 59.58 |
| **Age** | 18–24 years | 39 | **11.68** |
| | 25–34 years | 140 | **41.91** |
| | 35–44 years | 97 | **29.04** |
| | 45 and above | 58 | **17.36** |
| **Marital Status** | Single | 30 | **8.98** |
| | Married | 294 | **88.02** |
| | Divorced/Widowed | 10 | **2.99** |
| **Position** | Marketing Manager | 33 | **9.88** |
| | Accounting Manager | 110 | **59.88** |
| | Supervisors | 145 | **43.41** |
| | Operational Manager | 90 | **2.69** |
| | HR Manager | 8 | **2.39** |
| **Educational level** | Primary education | 10 | **2.99** |
| | Secondary education | 23 | **6.89** |
| | Diploma | 115 | **34.43** |
| | Bachelor Degree | 150 | **44.91** |
| | Master's Degree | 26 | **7.78** |
| | Ph.D. | 10 | **2.99** |
| **Organization work experience** | <4 years | 75 | **22.45** |
| | 4–6 | 68 | **20.36** |
| | 7–9 | 100 | **29.94** |
| | > 9years | 91 | **27.24** |

this survey, and only primary staff with years of experience in the field were selected. Micro-finance bankings were avoided in order to improve the generalization of the findings. Purposive quantitative research was employed for this study, and respondents were assured that their responses would be kept confidential. We contacted the supervisors and managers of each bank branch to seek their approval. In addition, 534 bank employees agreed to participate in the survey and, due to the current global situation, we mailed the questionnaires to the supervisors and managers. A total of 340 respondents responded to the survey, which represents a 63.7% response rate. A total of 6 incomplete questionnaires were deleted, and 334 valid questionnaires were used for the data analysis. The sample of the employees consists of 195 (59.58%) male and 139 (40.42%) female. A total of 41.91% were between 25 and 34 years of age, and 59.8% were supervisors with bachelor's degrees (44.91%). The banking employees' highest work experience is between 7–9 years at 29.94% as shown in Table 1.

## Measurement scales

*High-performance work practices* were measured using seven items adapted from previous literature [32]. The items focused on the key factors of high-performance work practices that have been extensively used in many kinds of literature, including training, empowerment, work-life balance, rewards, selective staffing, job security, and career opportunities [24, 25]. Employees were asked about the extent to which their organizations used high-performance work practices. The reliability test showed that the Cronbach's alpha value was 0.899, indicating acceptable reliability of the construct. *Career competencies* were measured using four items

adapted from Akkermans et al. [28]. Employees were asked to assess their career competencies in the organization. Cronbach's alpha was .879, indicates acceptable reliability of the construct. *Service quality* was measured using four items, adapted from Bettencourt et al. [39]. Service quality has been studied extensively [40–42]. The items include the quality of the service received within the organization. The overall reliability value was .879, indicating good reliability of the construct.

*Creative performance* was adapted from the scale developed by Ali and Razi [42]. In line with previous studies, Four items were used to assess the working environment, in line with previous studies [27, 43]. These items capture employees' perceptions of routine tasks in several ways. A reliability test using Cronbach's coefficient was conducted. Cronbach's alpha was 0.882, indicating acceptable reliability of the construct. *Extra-role performance* was measured and adapted from Ozturk and Karatepe [44]. The extra-role performance has been extensively studied [45–48]. Employees were asked to make constructive suggestions to improve the organization's operations. The Cronbach's coefficient was 0.869, indicating acceptable reliability of the construct.

The questionnaire was measured on a five-point Likert scale, ranging from strongly disagree (1) to strongly agree (5). We instructed bank employees to provide information on their high-performance work practices, career' competencies, creative performance, service quality, and extra-role performance. This ensured that this study adopted an approach to reduce common bias, as suggested by Weer and Greenhaus [49]. We ensured that respondents' responses were anonymous and confidential.

## Data analysis

This study carried out different tests to confirm the study proposed model. First, the model fit and other measurement factors were tested using CFA (Confirmatory Factor Analysis) as suggested by Podsakoff et al. [50] and Bentler [51]. Furthermore, we tested the research model relationships with Covariance-Based Structural Equation Modelling (CB-SEM) using the JASP 10.0.0.13 statistical software tool that measures relationships of direct and indirect effects of variables [52, 53].

## Results and findings

### Measurement model

The measurement model assesses the item's internal consistency and convergent validity. First, a preliminary test was performed. We checked the eigenvalues of components extracted. The analysis provided that some of the eigenvalues are above or equal to a value of one, as shown in Fig 2. The initial factor component (34%) of the total variance was explained. The eigenvalue component factors range from 1.263 to 7.146, and the percentage of total variance explained ranged from 34.8% to 71.6%. As a result, the variance of the common method does not reveal any serious problems in this study.

Furthermore, the results of skewness and kurtosis indices indicate that there is data normality of the model. As indicated, skewness and kurtosis indices should not exceed |2.3| to check for data normality. The value of skewness and kurtosis indices are between |2.3|. Hence, the respondent's data in this study indicate appropriateness for further factor analysis. In addition, Bartlett's test of sphericity was employed to measure the adequacy of the sampling and Kaiser–Meyer–Olkin (KMO). The results shown significant statistics of $\chi^2$ (210) = 4018.258 and the KMO measure = 0.879 > 0.500 respectively.

Second, we assess the measurement model based on internal consistency and convergent validity. The measurement model evaluates the factor loadings ($\gamma$), composite reliability (CR),

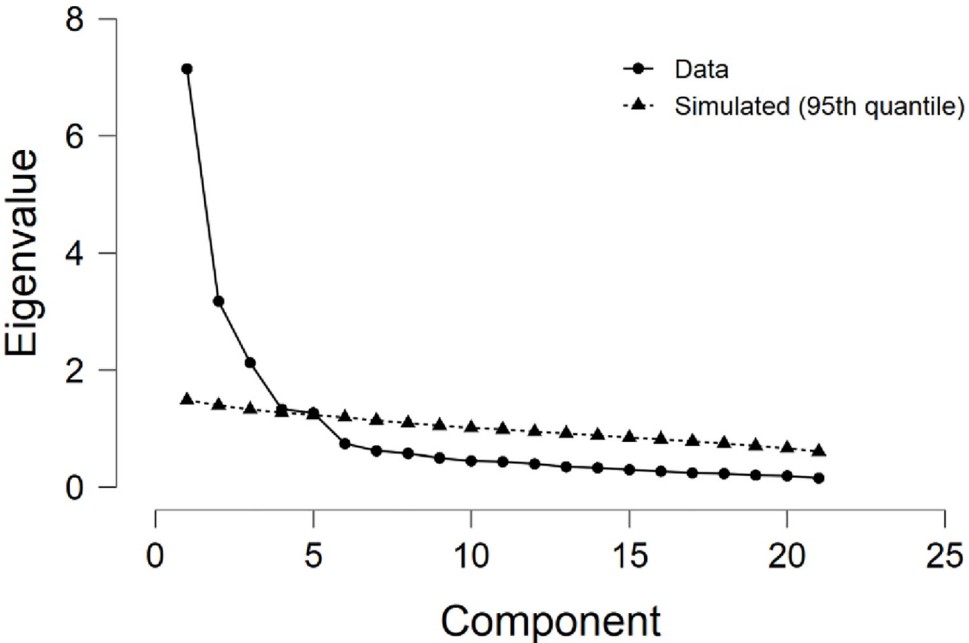

**Fig 2. Scree plots showing principal components and benchmark eigenvalue.**

average variance extracted (AVE), and Cronbach's alpha (α). As suggested by Han et al. [54], the recommended value for the reliability of each construct must be > 0.70. Therefore, the values of Cronbach's alpha (α) of the study ranged from 0.882 to 0.899, and values of each construct exceed 0.7 threshold value [54, 55], suggesting good internal consistency. Also, the factor loadings (γ), composite reliability (CR), and average variance extracted (AVE) were evaluated. The value of standard factor loading < 0.4 were deleted as recommended by [54]. Table 2 indicates that all values of factor loading are above 0.40. Also, the acceptable CR values for each construct should be > 0.70, while [54] recommended that the AVE values should also be > 0.50. The Cronbach's alpha and CR values exceed the 0.7 thresholds, and AVE exceeds the 0.5 threshold value. Thus, the value of standard factor loading, AVE, CR, and Cronbach's alpha satisfy the suggested thresholds. Table 2 presents a summary of the exploratory analyses (AVE, CR, Cronbach's alpha, skewness and kurtosis).

We conducted a correlation analysis to evaluate the relationships between the constructs [55]. Accordingly, the results show a positive and significant relationship between the constructs, as shown in Table 3. This implies that the correlation coefficients of the constructs satisfy the requirement acceptable for the structural model procedure. The mean, standard deviation, and bivariate correlations in Table 3 indicate a positive correlation between variables.

## The structural model

To further test the validity of the latent constructs, we used confirmatory factor analysis to confirm the model fit. Table 4 presents the results for the overall model fit. The chi-square/degree of freedom ($\chi^2$/df), Goodness of Fit Index (GFI), Bentler-Bonett Non-normed Fit Index (NNFI), Normed Fit Index (NFI), Root mean square error of approximation (RMSEA), Comparative Fit Index (CFI), and Standardized root mean square residual (SRMR) values were assessed based on range of recommended values [56, 57]. This results suggest that the research model is suitably fit.

**Table 2. Summary of exploratory analyses.**

| Construct | Factor Loading (λ) | Skewness | Kurtosis |
|---|---|---|---|
| **High-Performance Work Practices (HP)** | | | |
| **CR = 0.878, AVE = 0.590, α = 0.899** | | | |
| **HP1** | 0.735 | -1.17 | 0.808 |
| **HP2** | 0.799 | -1.14 | 1.39 |
| **HP3** | 0.844 | -1.12 | 1.20 |
| **HP4** | 0.702 | -0.735 | 0.095 |
| **HP5** | 0.753 | -1.38 | 1.67 |
| **Career' competencies (CC)** | | | |
| **CR = 0.799, AVE = 0.500, α = 0.879** | | | |
| **CC1** | 0.718 | -0.649 | -0.103 |
| **CC2** | 0.658 | -1.341 | -0.434 |
| **CC3** | 0.764 | -0.571 | -0.298 |
| **CC4** | 0.683 | -1.223 | -0.516 |
| **Service quality (SQ)** | | | |
| **CR = 0.814, AVE = 0.525, α = 0.790** | | | |
| **SQ1** | 0.739 | 0.132 | -0.820 |
| **SQ2** | 0.630 | 0.192 | -0.769 |
| **SQ3** | 0.811 | -0.096 | -1.798 |
| **SQ4** | 0.706 | -0.140 | -0.611 |
| **Creative performance (CP)**E | | | |
| **CR = 0.826, AVE = 0.544, α = 0.882** | | | |
| **CP1** | 0.700 | -0.766 | 0.198 |
| **CP2** | 0.702 | -1.20 | 1.430 |
| **CP3** | 0.776 | -0.469 | -0.194 |
| **CP4** | 0.768 | -0.385 | -0.374 |
| **Extra-role performance (ERP)** | | | |
| **CR = 0.842, AVE = 0.577, α = 0.869** | | | |
| **ERP1** | 0.795 | -0.943 | 0.882 |
| **ERP 2** | 0.916 | -0.950 | 0.546 |
| **ERP 3** | 0.710 | -1.34 | 1.86 |
| **ERP4** | 0.578 | -0.579 | -0.265 |

## Hypotheses tests

The structural method was tested using a bootstrapping procedure with 1000 bootstrap samples evaluated [58, 59]. The hypotheses tests show the relationship between the constructs for the research model shown in Table 5, and the path analysis presented the summary of results —the relationships, estimates of the original coefficients (β), Std. Error (standard error), z-value, and level of significance. Also, the $R^2$ for CC is 33.3%, which has proven to be significant for HP in line with [55]. Similarly, the predictive power of SQ (32.4%), ERP (10.5%), and CP (17.1%) proved to be significant for CC in the banking sector.

From the results, hypotheses H1, H2a, H2b, and H2c were empirically supported. Thus, the relationships among these hypotheses of the current research model were significant in predicting employee outcomes in banking sectors. Among these predictive factors, HP has a significant impact of (H1: β = 0.590, $p < 0.001$) on employee career competencies. This suggests that HP influence employee career competencies in the banking sector. In addition, employee career competencies significantly impact (H2a: β = 0.386, $\rho < 0.001$) on employee service quality in organization practices. This suggests that employee career competencies are an

**Table 3. Descriptive statistics and correlation coefficient of the constructs.**

| Constructs | Mean | SD | HP | CC | SQ | ERP | CP |
|---|---|---|---|---|---|---|---|
| HP | 3.822 | 0.128 | - | | | | |
| CC | 3.379 | 0.139 | 0.497 | - | | | |
| SQ | 3.715 | 0.179 | 0.310 | 0.485 | - | | |
| ERP | 3.827 | 2.242 | 0.258 | 0.289 | 0.133 | - | |
| CP | 2.848 | 0.149 | 0.265 | 0.354 | 0.233 | 0.561 | - |

* p < .05, ** p < .01, *** p < .001.

important factor that influences employee service quality in the banking sector. In addition, employees' career competencies have a positive effect (H2b: β = 0.288, ρ < 0.001) on creative performance in the banking sector. Thus, H3 was statistically significant. The findings indicate that employee career competencies have a significant effect (H2c: β = 0.354, ρ < 0.001) on extra-role performance in the banking sector. This suggests that employees' career competencies influence extra-role performance in the banking sector.

Also, the indirect effect of H3a shows the career competencies significantly influence a relationship between HP and service quality in the banking sector, and the result shows unverifiable support (H3a: β = 0.088, ρ < .0001), which indicates that employee' career competencies do not mediate the relationship between HP and service quality in the banking sector. In contrast, H3b shows that career competencies significantly influence the relationship between HP and creative performance. The results show empirical support (β = 0.114, ρ < .0001), which indicates that career competencies partially mediate the relationship between HP and creative performance in the banking sector. Also, H3c is hypothesized that career competencies significantly influence the relationship between HP and extra-role performance in banking sectors, the results show empirical support (β = 0.150, ρ > .0001) that career competencies partially mediates the relationship between HP and extra-role performance in the banking sector. Thus, a summary of the hypotheses test results is shown in Table 5.

## Discussion and implications

This study examines why HPWPs affect employees' career competencies. Further, we investigate whether employee career competencies influence employee service quality, creative performance, and employee extra-role performance in the banking sector.

First, there are some interesting findings regarding the effect of HPWPs on employee career competencies. In line with existing studies, the findings of this study provide empirical support for the direct relationship between HPWPs and employees' career competencies [24, 25]. The role of HPWPs tends to promote the importance of career-related competencies in areas such

**Table 4. Model fit summary of the research model (confirmatory factor analysis).**

| Fit index | Recommended value | Research model |
|---|---|---|
| Chi-square/degree of freedom ($\chi^2$/df) | $\leq$ 3.00 | 2.138 |
| Goodness of Fit Index (GFI) | $\geq$ 0.80 | 0.897 |
| Bentler-Bonett Non-normed Fit Index (NNFI) | $\geq$ 0.80 | 0.939 |
| Normed Fit Index (NFI) | $\geq$ 0.80 | 0.907 |
| Root mean square error of approximation (RMSEA) | $\leq$ 0.08 | 0.060 |
| Comparative Fit Index (CFI) | $\geq$ 0.90 | 0.948 |
| Tucker-Lewis Index (TLI) | $\geq$ 0.90 | 0.939 |
| Standardized root mean square residual (SRMR) | $\leq$ 0.05 | 0.041 |

**Table 5. Hypotheses test results.**

| Hypothesis | Estimate | Std. Error | z-value | p-value | Decision |
|---|---|---|---|---|---|
| **H1:HP→ CC** | 0.590 | 0.071 | 8.360 | 0.000 | Supported |
| **H2a:CC→ SQ** | 0.386 | 0.066 | 5.875 | 0.000 | Supported |
| **H2b:CC→ ERP** | 0.288 | 0.068 | 4.214 | 0.000 | Supported |
| **H2c:CC→ CP** | 0.354 | 0.068 | 5.196 | 0.000 | Supported |
| **H3a:HP→CC→ SQ** | 0.088 | 0.054 | 1.626 | 0.104 | Not supported |
| **H3b:HP→CC→ CP** | 0.114 | 0.058 | 1.972 | 0.049 | Supported |
| **H3c:HP→CC→ ERP** | 0.150 | 0.060 | 2.500 | 0.012 | Supported |

Supported at **$p < 0.05$; $z > 1.96$.

as knowledge, skills, and abilities through training, empowerment, and career opportunities. Accordingly, most organizations investing in HPWPs help staff's knowledge-intensive set of skills and competencies to help develop the relationships between HPWPs and employee career competencies and improve organizational core capabilities.

Second, this study's results demonstrated a significant relationship between career-related competencies and service quality. These results show that perceived employee career competencies have a positive impact on organizational service quality. This implies that employee professionals' roles and functions are determined by professional job activities being performed within the organization. It is a function of qualifying employees in performing certain functions and roles. Therefore, the link to employee competence in critical activities depends on the level of knowledge, skills, and attitude as it positively influences organizational service quality [34].

Third, this study's results revealed a significant effect between employee career competencies and creative performance, which is in line with prior studies [25, 37]. This shows that employees are willing to accomplish career goals and feel accomplished through selective staffing, training, and rewards to motivate their employees' creative performance and other social activities that might influence their creative performance. Therefore, career competencies help employees feel more secure and protected [60].

The findings of this study is in line with previous studies on career-related competencies and extra-role performance, further establishes that the relationship between career-related competencies and extra-role performance is significant to organizational practices [25, 48], in as much as the management is providing novel ideas to improve organizational outcomes when investing on HPWPs. Employees tend to enjoy extra-role performance as long as they fit their future career-related competencies. Thus, most employees tend to use their ability to develop an organization. Accordingly, when employees are motivated by organizational motivation, they find means of achieving their career goals with actions and new opportunities, they show loyalty through their extra-role performance capability.

Finally, this study confirmed the mediating effect of career competencies between HPWPs and creative performance, and the findings indicate career competencies are partially mediated. These results further indicate that strong career competencies strengthen the effect of HPWPs on organizational, creative performance. That is, an organization with HPWPs provides employees with creative performance to grow and develop the organization. However, the indirect relationship between HPWPs and service quality is not mediated by career competencies, suggesting an effect of employee career competence on the relationship between HPWPs and service quality. The result implies that employees who focus on career

competency do not strengthen organizational policies but only offer employees a long-term secure career for personal factors.

In conclusion, this study confirmed the mediating effect of career competencies on HPWPs and extra-role performance. The findings indicate that career competencies partially mediate the relationship between HPWPs and extra-role performance in the banking sector. These outcomes further indicate that strong career competencies strengthen the effect of HPWPs on employee extra-role performance. These results show that proper HPWPs help employees perform extra functions and roles at any level or position beyond organizational requirements. Thus, career competence with a high level of personal resources always accomplishes their exceptional goals in organizational development [61].

## Theoretical implications

From theoretical implications, this study contributes to organizational practices in the following ways. First, we highlight various studies on the effect of HPWSs on employee career competence and the subsequent employee outcomes—service quality, creative performance, and extra-role performance. Thus, this study provides a theoretical approach to understanding the mediating effect of career competency on the relationship between HPWPs and employee outcomes. This is important because no previous research has proposed a model to support this argument. This empirical study shows how HPWPs utilize their effects on employee career competence. In strategic human resource management studies, service quality, creative performance, and extra-role performance are imperative employee outcomes of career competencies. Thus, the analyses of these factors show a substantial contribution toward HR development in the banking sector.

In general, HPWPs effectively enhance employee career competence because they serve as organizational means for employees outcomes. Also, many researchers pay more attention to HPWSs as antecedents of different mediators to measure their effects on employee and organizational outcomes [25, 32]. Therefore, this study extends the study on the outcome of HPWPs by analyzing the boundary effect of employee career competence on the indirect relationship with employee outcome. Based on the results of employee outcomes, the indirect effect of service quality needs the attention of researchers on measures to improve organization/employee service quality.

## Practical implications

The practical implication of this study shows that investing in human capital is an asset to any organizational competitiveness and improving human resources, meaning that organizations need to maintain and retain their resources. Satisfied employees are more likely to stay with suitable staffing, general training and development, incentive rewards, good work-life balance, adequate empowerment, job security, and excellent career opportunities. It determines employees' well-being and organizational performance [8, 9]. Therefore, the findings imply that organizations can achieve employee outcomes by implementing proper HPW practices that help develop and manage the interrelationships among employees in any organization.

In addition, bank managers and supervisors need to focus on HPWPs to develop employees' career competencies because, without human capital development, the organization might be affected internally and externally. This is important because employees' career competency promotes a positive environment in every organization and improves employee outcomes. In retrospect, managers should provide incentives, royalties, and performance activities so that there will be an effective means of communication, knowledge sharing, and goal targets within the organization.

Moreover, the findings of this study suggest that managers should develop more elusive career competence programs that will benefit from the organizational application of high-performance work practices in workplaces. In this respect, bank managers and supervisors need to improve employees' career competencies to promote strategic human resource management and help reduce incompetence within organizations.

## Limitations and suggestions for future research

This study has many limitations; and we suggest better future research. First, this study used a single-time cross-sectional approach with many advantages and significant contributions to providing vital information for the study. However, a longitudinal approach could be used for future studies to track trends and changes over time with the same respondents. Furthermore, the inclusion of other control variables such as, age and gender differences could be an additional measure for future research on HPW practices.

Second, the present study focuses specifically on banks in Tanzania with respect to HR policies and practices in their organizations. Thus, there is a need to extend the study population; multiple locations and other non-financial institutions to provide a better concept for future research. This study measured the effect of high-performance work practices on employee career competencies on service quality, creative performance, and extra-role performance. However, the study should consider in-depth aspects of HPWPs such as training, empowerment, work-life balance, rewards, selective staffing, job security, and career opportunities to provide insights into the current study. Therefore, Therefore, there is a need to expand the study to a more comprehensive aspect of HPWPs in order to provide a better concept for future research.

Finally, the current study examines how high-performance work practices influence employee career competencies and their influences on service quality, creative performance, and extra-role performance as a resultant. Future research should take into account the management and policies of other non-financial institutions which may assist in decision-making and organizational growth.

## Supporting information

**S1 File.**
(DOCX)

## Author Contributions

**Conceptualization:** Damis Feruzi Kamna.

**Formal analysis:** Shiva Ilkhanizadeh.

**Investigation:** Damis Feruzi Kamna.

**Methodology:** Damis Feruzi Kamna.

**Project administration:** Shiva Ilkhanizadeh.

**Supervision:** Shiva Ilkhanizadeh.

**Validation:** Shiva Ilkhanizadeh.

**Writing – original draft:** Damis Feruzi Kamna.

**Writing – review & editing:** Shiva Ilkhanizadeh.

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
