## [Decision Letter · Decision Letter 0]

20 Aug 2021

PONE-D-21-22454

CAN HIGH PERFORMANCE WORK PRACTICES HAVE EFFECT ON EMPLOYEE CAREER COMPETENCIES? THERE IS A NEED TO IMPROVE EMPLOYEE OUTCOMES.

PLOS ONE

Dear Dr. Damis Feruzi Kamna

Thank you for submitting your manuscript to PLOS ONE. After careful consideration, we feel that it has merit but does not fully meet PLOS ONE’s publication criteria as it currently stands. Therefore, we invite you to submit a revised version of the manuscript that addresses the points raised during the review process.

We look forward to receiving your revised manuscript.

Kind regards,

Yuriy Bilan

Academic Editor

PLOS ONE

Journal Requirements:

6. Please amend your authorship list in your manuscript file to include author Shiva ILKHANIZADEH and Damis Feruzi Kamna.

Additional Editor Comments (if provided):

Major revision is needed

Reviewers' comments:

Reviewer's Responses to Questions

**Comments to the Author**

1. Is the manuscript technically sound, and do the data support the conclusions?

Reviewer #1: Partly

Reviewer #2: Yes

2. Has the statistical analysis been performed appropriately and rigorously? 

Reviewer #1: No

Reviewer #2: Yes

3. Have the authors made all data underlying the findings in their manuscript fully available?

Reviewer #1: No

Reviewer #2: Yes

4. Is the manuscript presented in an intelligible fashion and written in standard English?

Reviewer #1: Yes

Reviewer #2: Yes

5. Review Comments to the Author

Reviewer #1: 1. It is not clear whether the findings of the study can be disseminated to all Tanzanian bank employees, as the representativeness of the sample (340 respondents) using the Cochran formula (Cochran, 1977) is not justified.

such as https://doi.org/10.3846/btp.2020.12750

2. The Measurement Scales section discusses 5 items of High Performance Work Practices, 4 items of Career’s competencies, 4 items of Service quality, 4 items of Creative performance, 4 items of Extra-role performance. They are also listed in table 2: CP-CP5, CC1-CC4, SQ1-SQ4, CP1-CP4 (coincide with the first group of CP-CP5), ERP1-ERP4. Also in table 3. However, in the text of the article they are not explained.

3. It is also not clear what is the essence of the adaptation of the literature [31], [27], [38], [42], [43], as stated in the section Measurement Scales. For a description of High Performance Work Practices you can for example use https://doi.org/10.3846/btp.2020.12750

4. It is not clear which component is referred to in Figure 2., in what units it is measured

5. It is necessary to specify Measurement in table 1

6. Table 3 is called "Correlation coefficient of the constructs". But there are still Mean, SD. Therefore it is necessary to specify the name or the maintenance of table 3.

7. Is it correct that in table 3 the correlation coefficient of, for example, HPWPs with HPWPs = 0.768, and not 1,000? This also applies to other diagonal correlation coefficients

8. Figure 1. Conceptual model - in the title of the figure you need to specify which model it is

9. Non-content errors. The study examined the effect of high performance work practices on employee outcomes in banking system - repetition of words in the Abstract section. Are De sala salaam, Arub, Nwanza cities in Tanzania? are they spelled correctly?

Reviewer #2: The study was conducted on the basis of the banking sector. This aspect should be reflected in the title of the article.

It is also worth mentioning modern concepts that contain elements of HR.

References need updating because there are outdated sources.

Other comments are in the text.

6. PLOS authors have the option to publish the peer review history of their article (what does this mean?). If published, this will include your full peer review and any attached files.

Reviewer #1: No

Reviewer #2: No

---

## [Author Response · Author response to Decision Letter 0]

15 Sep 2021

Response to reviewers

Manuscript ID PONE-D-21-22454

We've checked your submission and before we can proceed, we need you to address the following issues:

1.  Thank you for stating the following financial disclosure:

4. Please amend your authorship list in your manuscript file to include author Shiva ILKHANIZADEH and Damis Feruzi Kamna.

5.Please provide additional details regarding participant consent. In the Methods section, please ensure that you have specified (1) whether consent was informed and (2) what type you obtained (for instance, written or verbal). If your study included minors, state whether you obtained consent from parents or guardians. If the need for consent was waived by the ethics committee, please include this information.

6. We suggest you thoroughly copyedit your manuscript for language usage, spelling, and grammar. If you do not know anyone who can help you do this, you may wish to consider employing a professional scientific editing service.

7. Please ensure that you refer to Table 1 in your text as, if accepted, production will need this reference to link the reader to the Table.

Your comments are appreciated. We have added and met all the editorial requirements for 1-8.

Response to reviewers

 Authors comment Corrections made

Reviewer #1: 1. It is not clear whether the findings of the study can be disseminated to all Tanzanian bank employees, as the representativeness of the sample (340 respondents) using the Cochran formula (Cochran, 1977) is not justified. We thank you for the constructive reviews. we justified the data through secondary data and Yamane sample size calculator. Data from this survey were collected through online questionnaire conducted with banking employees across Tanzania - Dar es Salaam, Arusha, Mwanza, Moshi and Dodoma. Dar es Salaam has 290 branches, followed by Arusha, 68 branches, Mwanza, 67, Moshi, 46, and Dodoma, 41. Available data from Bank of Tanzania annual report shows that in 2020, banking supervisors and managers in this region have a total of 1,198 employees. Therefore, the Yamane (1967) sample size calculator was used to establish the minimum sample size for the survey and the result shows that 301 participants are required for the survey. 

The Measurement Scales section discusses 5 items of High Performance Work Practices, 4 items of Career’s competencies, 4 items of Service quality, 4 items of Creative performance, 4 items of Extra-role performance. They are also listed in table 2: CP-CP5, CC1-CC4, SQ1-SQ4, CP1-CP4 (coincide with the first group of CP-CP5), ERP1-ERP4. Also in table 3. However, in the text of the article they are not explained. We thank you for the feedbacks. We corrected High Performance Work Practices and all other issues with the segment thanks High Performance Work Practices (HP);

Career’ competencies (CC)

Service quality (SQ)

Creative performance (CP)

Extra-role performance (ERP)

It is also not clear what is the essence of the adaptation of the literature [31], [27], [38], [42], [43], as stated in the section Measurement Scales. For a description of High Performance Work Practices you can for example use https://doi.org/10.3846/btp.2020.12750 Thanks for your observation. We have to adopt it from different literature.

Thnks It is because we adapted from different literature instead one literature 

It is not clear which component is referred to in Figure 2., in what units it is measured Thanks for your observation. We have corrected Figure 2. Scree plots showing principal components and benchmark Eigenvalue

It is necessary to specify Measurement in table 1 Thanks for the observation Thanks we removed the measurement as you have rightly stated

Table 3 is called "Correlation coefficient of the constructs". But there are still Mean, SD. Therefore it is necessary to specify the name or the maintenance of table 3. We thank you for the constructive review. we have made the corrections. Table 3 Descriptive statistics and correlation coefficient of the constructs

Figure 1. Conceptual model - in the title of the figure you need to specify which model it is We thank you for your constructive review. we made the adjustments. The research model of this study is depicted in Figure 1. based on the effect of high performance work practices on employee career competencies and the mediating effect of employee career competencies on the relationship between high performance work practices and employee outcomes in the banking sector

Reviewer 2 

Non-content errors. The study examined the effect of high performance work practices on employee outcomes in banking system - repetition of words in the Abstract section. Are De sala salaam, Arub, Nwanza cities in Tanzania? are they spelled correctly? We thank you for your constructive review. we made the adjustments. In today's world, it is important for organisations to invest and improve employee outcomes to enhance organizational competitiveness and growth. However, most organizations place management objectives above the career competencies of employees. Therefore, this study investigates 1. the effect of high performance work practices on employee career competencies in the banking industry. 2. the mediating effect of employee career competencies on the relationship between high performance work practices and employee outcomes in the banking sector

Dar es Salaam, Arusha, Mwanza, Moshi and Dodoma

Other Corrections Thanks for the observation CAN HIGH PERFORMANCE WORK PRACTICES INFLUENCE EMPLOYEE CAREER COMPETENCIES ? THERE IS A NEED FOR BETTER EMPLOYEE OUTCOMES IN THE BANKING INDUSTRY.

It is also worth mentioning modern concepts that contain elements of HR. For example, Social responsibility assessment in the field of employment (case study of manufacturing). doi.org/10.33271/nvngu/2020-3/131 We thank you for your constructive review. we made the adjustments. It is necessary for an organization to select and design the appropriate HR practices that are not only allied with each other, but also affiliated with the organization's plan so as to accomplish the suitable organizational outcomes [1,10] such as several components of HPWSs and the assessment of social responsibility in employment relations [24-26].

This approach also has other positive consequences for organizations. For example, Hiring and retaining skilled employees in SMEs: problems in human resource practices and links with organizational success. We appreciate your comments. we have corrected it. With HPWPs in place, employees relish working for the organization while following their objectives and building their career competencies. This occurs when organizations carefully recruit the appropriate personnel to boost their productivity. According to [31], employee empowerment promotes competence, growth and helps employees reach their goals as long as employees continue to receive training adapted to their qualifications. This increases their capacity and provides them with the opportunity to use their empowerment in their work successfully. Also, encourage them to effectively manage their job requirements while working in an environment where remuneration, career development and job security are highly considered [32]. As previously studied by [25] and also stated that HPWPs has a significant relationship with career adaptability. Therefore, this hypothesis is hereby adopted: 

References need updating because there are outdated sources (1991, 1995, 1996, 1998…) Thanks for the observation We updated the references

---

## [Decision Letter · Decision Letter 1]

5 Jan 2022

PONE-D-21-22454R1CAN HIGH PERFORMANCE WORK PRACTICES INFLUENCE EMPLOYEE CAREER COMPETENCIES ? THERE IS A NEED FOR BETTER EMPLOYEE OUTCOMES IN THE BANKING INDUSTRY.PLOS ONE

Dear Dr. Damis Feruzi Kamna,

Thank you for submitting your manuscript to PLOS ONE. After careful consideration, we feel that it has merit but does not fully meet PLOS ONE’s publication criteria as it currently stands. Therefore, we invite you to submit a revised version of the manuscript that addresses the points raised during the review process.

ACADEMIC EDITOR:

The manuscript needs to be reviewed by an editing service. A native English speaker should proofread the writing.

In addition, authors should correct the following issues:

The acronym HPWPs appears in the abstract, but the reader does not know what it is since it has not yet been described. Therefore, delete and write the full meaning.Line 43: “[8] also contended…” should be changed by “Lepack et al. [8] also contended…”Analogously, in line 48: “Szabó-Bálint [19] described a career…”Line 62: “than examines High…” should be changed by “than examines high…”HPWPs has been not described previously. It could be done on line 58.Line 101: “According to [31]…” should be changed by “According to Kong et al. [31]…”Lines 107-108: “As previously studied by [26] and…” should be changed by “As previously studied by Oliinyk [26] and…”Line 126: “Creative performance the creation of…” The sentence is not complete.Line 144 should be in bold.Lines 166-167: What are Dar es Salaam, Arusha, Mwanza, Moshi and Dodoma. They are banks? It should be specified.Line 170: What sample calculation criteria were used?Line 183: “With highest age between 25-34 years…”. I suggest to change the sentence because it lead to think that the older employees had between 25-34 years. In reality, the largest group of employees were between 25-34 years old.Line 184: Change “Bachelor Degree” by “bachelor degree”.Lines 190-194: Authors say that HPWP were measured using 5 items. However, training, empowerment, work-life balance, rewards, selective staffing, job security and career opportunities are 7 items.Line 199: “… adapted from [28]” should be changed by “… adapted from Akkermans et al. [28]”. Idem in lines 204, 210 and 218Lines 199-203: What are those 4 items? They should be specified. Analogously, lines 204-209, 210-217 and 218-228.Line 212: Change “we” by “We”.Line 227: Change “… as suggested by [49]” by “… as suggested by Weer and Greenhaus [49]”All paragraphs in the "Measurement scales" section are constructed in the same way. It is not necessary to indicate 5 times “five-point Likert scale, ranging from strongly disagree (1) to strongly agree (5)". Once is enough since it is the same in all items. Rewrite this section combining concepts and not repeating the wording scheme.Revise line 236: “… as required by [50,51]”Rewrite sentence in line 243.Revise sentence in line 245-246.Revise the present-past in lines 248-249.PLS-SEM do not requires data be normal distribution. Therefore, the analysis of normality has no sense in this study (lines 251-261).Revise sentence in line 262: “we measured the measurement model…”. Why not, for example, “we assess the measurement model…”. In line 263 authors repeat “measures”. Please, improve the wording.Revise line 270: “… as recommended by [54]”Line 275. “As presented in Table 2” is an incomplete sentence.Lines 279-280 are not clear.Line 307: Revise “as stated by [55]”Lines 309-310: It is not correct. Authors should revise this statement. SQ, ERP and CP are not predictors of CC.Line 314 onwards HPWP has been changed by HPW. Why?Lines 311-325 has been constructed in a similar way. Please, change the wording.==============================

Please submit your revised manuscript by Feb 19 2022 11:59PM If you will need more time than this to complete your revisions, please reply to this message or contact the journal office at plosone@plos.org. Please include the following items when submitting your revised manuscript:A rebuttal letter that responds to each point raised by the academic editor and reviewer(s). You should upload this letter as a separate file labeled 'Response to Reviewers'.A marked-up copy of your manuscript that highlights changes made to the original version. You should upload this as a separate file labeled 'Revised Manuscript with Track Changes'.An unmarked version of your revised paper without tracked changes. You should upload this as a separate file labeled 'Manuscript'.If applicable, we recommend that you deposit your laboratory protocols in protocols.io to enhance the reproducibility of your results. Protocols.io assigns your protocol its own identifier (DOI) so that it can be cited independently in the future. For instructions see: https://journals.plos.org/plosone/s/submission-guidelines#loc-laboratory-protocols. Additionally, PLOS ONE offers an option for publishing peer-reviewed Lab Protocol articles, which describe protocols hosted on protocols.io. Read more information on sharing protocols at https://plos.org/protocols?utm_medium=editorial-email&utm_source=authorletters&utm_campaign=protocols.

We look forward to receiving your revised manuscript.

Kind regards,

María del Carmen Valls Martínez, Ph.D.

Academic Editor

PLOS ONE

Journal Requirements:

Reviewers' comments:

Reviewer's Responses to Questions

**Comments to the Author**

1. If the authors have adequately addressed your comments raised in a previous round of review and you feel that this manuscript is now acceptable for publication, you may indicate that here to bypass the “Comments to the Author” section, enter your conflict of interest statement in the “Confidential to Editor” section, and submit your "Accept" recommendation.

Reviewer #2: (No Response)

2. Is the manuscript technically sound, and do the data support the conclusions?

Reviewer #2: Yes

3. Has the statistical analysis been performed appropriately and rigorously? 

Reviewer #2: Yes

4. Have the authors made all data underlying the findings in their manuscript fully available?

Reviewer #2: Yes

5. Is the manuscript presented in an intelligible fashion and written in standard English?

Reviewer #2: Yes

6. Review Comments to the Author

Reviewer #2: (No Response)

7. PLOS authors have the option to publish the peer review history of their article (what does this mean?). If published, this will include your full peer review and any attached files.

Reviewer #2: No

---

## [Author Response · Author response to Decision Letter 1]

31 Jan 2022

Dear EIC/reviewers

We will like to appreciate your efforts in reviewing this manuscript.

Thank you for your comments and contributions, it helps the improve the readability of the manuscript.

Thanks.

The acronym HPWPs appears in the abstract, but the reader does not know what it is since it has not yet been described. Therefore, delete and write the full meaning.

Thanks for the comment

We have deleted it and added the full meaning of the acronyms

Thanks

Line 43: “[8] also contended…” should be changed by “Lepack et al. [8] also contended…”

Thanks for the comment

We have changed it to [8] also contended 

Analogously, in line 48: “Szabó-Bálint [19] described a career…”

Thanks for the comment

We have changed it to [19] described a career

Line 62: “than examines High…” should be changed by “than examines high…” 

Thanks for the comment

We corrected it - than examines high

HPWPs has been not described previously. It could be done on line 58.

Thanks for the comment

We made proper correction - high-performance work practices (HPWPs)

Thanks

Line 101: “According to [31]…” should be changed by “According to Kong et al. [31]…”

Thanks for the comment

We have changed it to According to [31] 

Lines 107-108: “As previously studied by [26] and…” should be changed by “As previously studied by Oliinyk [26] and…”

Thanks for the comment

We have changed it to According to [26]

Line 126: “Creative performance the creation of…” The sentence is not complete.

Thanks for the comment

We have changed it to - Creative performance is the creation that influences “products, ideas, or procedures that satisfy two conditions, namely, they are novel or original, and they are potentially relevant for, or useful to an organization”

Line 144 should be in bold.

Thanks for the comment

Yes we bold it - competencies as mediators

Lines 166-167: What are Dar es Salaam, Arusha, Mwanza, Moshi and Dodoma. They are banks? It should be specified.

Thanks for the comment

Yes we specified the main cities where the data collection took place- main cities of Dar es Salaam, Arusha, Mwanza, Moshi, and Dodoma

Line 170: What sample calculation criteria were used?

Thanks for the comment

Yes, we used (yamane, 1967) calculator n/1+n(e2). Yamane T. Statistics, an introductory Analysis 2nd Edition: Harper and Row. New York. 1967.

Line 183: “With highest age between 25-34 years…”. I suggest to change the sentence because it lead to think that the older employees had between 25-34 years. In reality, the largest group of employees were between 25-34 years old.

A total of 41.91% were between 25 and 34 years of age, and 59.8% were supervisors with bachelor’s degrees (44.91%). 

Thanks for the comment

Line 184: Change “Bachelor Degree” by “bachelor degree”.

Thanks for the comment

We have changed it 

Lines 190-194: Authors say that HPWP were measured using 5 items. However, training, empowerment, work-life balance, rewards, selective staffing, job security and career opportunities are 7 items.

Thanks for the comment

We have corrected it 

Line 199: “… adapted from [28]” should be changed by “… adapted from Akkermans et al. [28]”. Idem in lines 204, 210 and 218

Thanks for the comment

We have corrected it 

Line 212: Change “we” by “We”.

Thanks for the comment - This study tested for the analysis of descriptive statistics

Line 227: Change “… as suggested by [49]” by “… as suggested by Weer and Greenhaus [49]”

Thanks for the comment- We have corrected it 

All paragraphs in the "Measurement scales" section are constructed in the same way. It is not necessary to indicate 5 times “five-point Likert scale, ranging from strongly disagree (1) to strongly agree (5)". Once is enough since it is the same in all items. Rewrite this section combining concepts and not repeating the wording scheme.

Thanks for the comment- We have corrected it 

Revise line 236: “… as required by [50,51]”

Rewrite sentence in line 243.

Revise sentence in line 245-246.

Revise the present-past in lines 248-249.

Thanks for the comment- We have corrected it 

This study carried out different tests to confirm the study proposed model. This study tested for the analysis of descriptive statistics. The model fit and other factor measures were checked through confirmatory factor analysis (CFA), through measurement model. We tested the whole model in CFA consist of five constructs, and the study checked for the model fit indices as suggested by [50,51]. Also, we tested the research model fit with structural equation modeling using the JASP 10.0.0.13 statistical software tool that measures relationships of direct and indirect effects in relation to previous studies [52,53]. 

PLS-SEM do not requires data be normal distribution. Therefore, the analysis of normality has no sense in this study (lines 251-261).

Thanks for the comment- We have corrected it - Covariance Based Structural Equation Modelling (CB-SEM).

Revise sentence in line 262: “we measured the measurement model…”. Why not, for example, “we assess the measurement model…”. In line 263 authors repeat “measures”. Please, improve the wording.

Revise line 270: “… as recommended by [54]”

Line 275. “As presented in Table 2” is an incomplete sentence.

Lines 279-280 are not clear.

Line 307: Revise “as stated by [55]”

Thanks for the comment- We have corrected this section

Second, we assess the measurement model based on internal consistency and convergent validity. The measurement model evaluates the factor loadings (γ), composite reliability (CR), average variance extracted (AVE), and Cronbach’s alpha (α). As suggested [54], the recommended value for the reliability of each construct must be > 0.70. Therefore, the values of Cronbach’s alpha (α) of the study ranged from 0.882 to 0.899, and values of each construct exceed 0.7 threshold value [54,55], suggesting good internal consistency. Also, the factor loadings (γ), composite reliability (CR), and average variance extracted (AVE) were evaluated. The value of standard factor loading < 0.4 were deleted as recommended by [54]. Table 2 indicates that all values of factor loading are above 0.40. Also, the acceptable CR values for each construct should be > 0.70, while the AVE values should also be > 0.50, as recommended by [54]. The Cronbach's alpha and CR values exceed the 0.7 thresholds, and AVE exceeds the 0.5 threshold value. Thus, the value of standard factor loading, AVE, CR, and Cronbach's alpha satisfy the suggested thresholds. Table 2 presents a summary of the exploratory analyses (AVE, CR, Cronbach's alpha, skewness and kurtosis ). 

Lines 309-310: It is not correct. Authors should revise this statement. 

SQ, ERP and CP are not predictors of CC.

Also, the R2 for CC is 33.3%, which has proven to be significant for HP in line with [55]. Similarly, the predictive power of SQ (32.4%), ERP (10.5%), and CP (17.1%) proved to be significant for CC in the banking sector.

Line 314 onwards HPWP has been changed by HPW. Why?

Lines 311-325 has been constructed in a similar way. Please, change the wording.

Thanks for the comment- We have corrected it 

Thnaks

---

## [Editor Report · Decision Letter 2]

10 Feb 2022

PONE-D-21-22454R2CAN HIGH PERFORMANCE WORK PRACTICES INFLUENCE EMPLOYEE CAREER COMPETENCIES ? THERE IS A NEED FOR BETTER EMPLOYEE OUTCOMES IN THE BANKING INDUSTRY.PLOS ONE

Dear Dr. Damis Feruzi Kamna,

Thank you for submitting your manuscript to PLOS ONE. After careful consideration, we feel that it has merit but does not fully meet PLOS ONE’s publication criteria as it currently stands. Therefore, we invite you to submit a revised version of the manuscript that addresses the points raised during the review process.

The manuscript has not been reviewed by a native English speaker, as requested. Please send the manuscript to an editing service.

The changes indicated in the previous letter were also not made. They must be made:

• Line 43: “[8] also contended…” should be changed by “Lepack et al. [8] also contended…”

• Analogously, in line 48: “Szabó-Bálint [19] described a career…”

• Line 101: “According to [31]…” should be changed by “According to Kong et al. [31]…”

• Lines 107-108: “As previously studied by [26] and…” should be changed by “As previously studied by Oliinyk [26] and…”

• Line 170: What sample calculation criteria were used? You must mention it in the text.

• Line 199: “… adapted from [28]” should be changed by “… adapted from Akkermans et al. [28]”. Idem in lines 204, 210 and 218

• Lines 199-203: What are those 4 items? They should be specified. Analogously, lines 204-209, 210-217 and 218-228.

• Line 212: Change “we” by “We”.

• Line 227: Change “… as suggested by [49]” by “… as suggested by Weer and Greenhaus [49]”

• All paragraphs in the "Measurement scales" section are constructed in the same way. It is not necessary to indicate 5 times “five-point Likert scale, ranging from strongly disagree (1) to strongly agree (5)". Once is enough since it is the same in all items. Rewrite this section combining concepts and not repeating the wording scheme.

• Revise line 236: “… as required by [50,51]”

• Rewrite sentence in line 243.

• Revise the present-past in lines 248-249.

• PLS-SEM do not requires data be normal distribution. Therefore, the analysis of normality has no sense in this study (lines 251-261).

• Revise line 270: “… as recommended by [54]”

• Line 307: Revise “as stated by [55]”

We look forward to receiving your revised manuscript.

Kind regards,

María del Carmen Valls Martínez, Ph.D.

Academic Editor

PLOS ONE
---

## [Author Response · Author response to Decision Letter 2]

14 Feb 2022

Dear EIC/reviewers

We will like to appreciate your efforts in reviewing this manuscript.

Thank you for your comments and contributions, it helps improve the readability of the manuscript.

Thanks.

Line 43: “[8] also contended…” should be changed by “Lepack et al. [8] also contended…”

Thanks for the comment

We have changed it to Lepack et al. [8] also stated 

Analogously, in line 48: “Szabó-Bálint [19] described a career…”

Thanks for the comment

We have changed it to Szabó-Bálint [19] described a career

Line 101: “According to [31]…” should be changed by “According to Kong et al. [31]…”

Thanks for the comment

We have changed it to According to Kong et al. [31],

Lines 107-108: “As previously studied by [26] and…” should be changed by “As previously studied by Oliinyk [26] and…”

Thanks for the comment

We has been changed to Oliinyk [26] stated 

Line 170: What sample calculation criteria were used?

Thanks for the comment

Yes, we used Yamane [38] sample size calculator was used to establish the minimum sample size for the survey, and the result shows that 301 participants are required for the survey. 

Line 199: “… adapted from [28]” should be changed by “… adapted from Akkermans et al. [28]”. Idem in lines 204, 210 and 218

Thanks for the comment

We have corrected it 

Line 212: Change “we” by “We”.

Thanks for the comment - This study tested for the analysis of descriptive statistics

Line 227: Change “… as suggested by [49]” by “… as suggested by Weer and Greenhaus [49]”

Thanks for the comment- We have corrected it 

This ensured that this study adopted an approach to reduce common bias, as suggested by Weer and Greenhaus [49].

All paragraphs in the "Measurement scales" section are constructed in the same way. It is not necessary to indicate 5 times “five-point Likert scale, ranging from strongly disagree (1) to strongly agree (5)". Once is enough since it is the same in all items. Rewrite this section combining concepts and not repeating the wording scheme.

Thanks for the comment- We have corrected it 

High-performance work practices were measured using seven items adapted from previous literature [32]. The items focused on the key factors of high-performance work practices that have been extensively used in many kinds of literature, including training, empowerment, work-life balance, rewards, selective staffing, job security, and career opportunities [24,25]. Employees were asked about the extent to which their organizations used high-performance work practices. The reliability test showed that the Cronbach’s alpha value was 0.899, indicating acceptable reliability of the construct. Career competencies were measured using four items adapted from Akkermans et al. [28]. Employees were asked to assess their career competencies in the organization. Cronbach’s alpha was .879, indicates acceptable reliability of the construct. Service quality was measured using four items, adapted from Bettencourt et al. [39]. Service quality has been studied extensively [40-42]. The items include the quality of the service received within the organization. The overall reliability value was .879, indicating good reliability of the construct. 

Creative performance was adapted from the scale developed by Ali and Razi [42]. In line with previous studies, Four items were used to assess the working environment, in line with previous studies [27,43]. These items capture employees’ perceptions of routine tasks in several ways. A reliability test using Cronbach’s coefficient was conducted. Cronbach’s alpha was 0.882, indicating acceptable reliability of the construct. Extra-role performance was measured and adapted from Ozturk and Karatepe [44]. The extra-role performance has been extensively studied [45-48]. Employees were asked to make constructive suggestions to improve the organization's operations. The Cronbach’s coefficient was 0.869, indicating acceptable reliability of the construct. The questionnaire was measured on a five-point Likert scale, ranging from strongly disagree (1) to strongly agree (5). We instructed bank employees to provide information on their high-performance work practices, career’ competencies, creative performance, service quality, and extra-role performance. This ensured that this study adopted an approach to reduce common bias, as suggested by Weer and Greenhaus [49]. We ensured that respondents' responses were anonymous and confidential.

Revise line 236: “… as required by [50,51]”

Rewrite sentence in line 243.

Revise sentence in line 245-246.

Revise the present-past in lines 248-249.

Thanks for the comments- We have corrected it 

This study carried out different tests to confirm the study proposed model. First, the model fit and other measurement factors were tested using CFA (Confirmatory Factor Analysis) as suggested by Podsakoff et al. [50] and Bentler [51]. Furthermore, we tested the research model relationships with Covariance-Based Structural Equation Modelling (CB-SEM) using the JASP 10.0.0.13 statistical software tool that measures relationships of direct and indirect effects of variables [52,53]. 

PLS-SEM do not requires data be normal distribution. Therefore, the analysis of normality has no sense in this study (lines 251-261).

Thanks for the comment- We have corrected it - we used Covariance Based Structural Equation Modelling (CB-SEM).

RRevise line 270: “… as recommended by [54]”

while [54] recommended that the AVE values should also be > 0.50.

Line 275. “As presented in Table 2” is an incomplete sentence.

Table 2 presents a summary of the exploratory analyses (AVE, CR, Cronbach's alpha, skewness and kurtosis). 

Line 307: Revise “as stated by [55]”

Thanks for the comment- We have corrected this section

We conducted a correlation analysis to evaluate the relationships between the constructs [55]. Accordingly, the results show a positive and significant relationship between the constructs

---

## [Editor Report · Decision Letter 3]

17 Feb 2022

CAN HIGH PERFORMANCE WORK PRACTICES INFLUENCE EMPLOYEE CAREER COMPETENCIES ? THERE IS A NEED FOR BETTER EMPLOYEE OUTCOMES IN THE BANKING INDUSTRY.

PONE-D-21-22454R3

Dear Dr. Damis Feruzi Kamna,

We’re pleased to inform you that your manuscript has been judged scientifically suitable for publication and will be formally accepted for publication once it meets all outstanding technical requirements.

Kind regards,

María del Carmen Valls Martínez, Ph.D.

Academic Editor

PLOS ONE
---

## [Editor Report · Acceptance letter]

21 Feb 2022

PONE-D-21-22454R3 

CAN HIGH-PERFORMANCE WORK PRACTICES INFLUENCE EMPLOYEE CAREER COMPETENCIES? THERE IS A NEED FOR BETTER EMPLOYEE OUTCOMES IN THE BANKING INDUSTRY. 

Dear Dr. Kamna:

I'm pleased to inform you that your manuscript has been deemed suitable for publication in PLOS ONE. Congratulations! Your manuscript is now with our production department. 

Kind regards, 

on behalf of

Dr. María del Carmen Valls Martínez 

Academic Editor

PLOS ONE